# Motor control exercises versus general exercises for greater trochanteric pain syndrome: A protocol of a randomized controlled trial

**Guilherme Thomaz de Aquino Nava** [1]*, **Caroline Baldini Prudencio** [2], **Rafael Krasic Alaiti** [3], **Beatriz Mendes Tozim** [4], **Rebecca Mellor** [5], **Cristiane Rodrigues Pedroni** [1,4], **Angélica Mércia Pascon Barbosa** [2,4], **Marcelo Tavella Navega** [1,4]

1 Department of Physical Education, Institute of Biosciences, São Paulo State University (UNESP), Rio Claro, São Paulo, Brazil, 2 Department of Gynecology and Obstetrics, Botucatu Medical School, São Paulo State University (UNESP), Botucatu, São Paulo, Brazil, 3 Nucleus of Neuroscience and Behavior and Nucleus of Applied Neuroscience, Universidade de São Paulo (USP), São Paulo, Brazil, 4 Department of Physiotherapy and Occupational Therapy, Faculty of Philosophy and Sciences, São Paulo State University (UNESP), Marilia, São Paulo, Brazil, 5 School of Health and Rehabilitation Sciences, The University of Queensland, Brisbane, Queensland, Australia

☯ These authors contributed equally to this work.

* gtanava@gmail.com

## Abstract

## Introduction

Greater trochanteric pain syndrome is an overarching term used to define pain and tenderness in the greater trochanteric region of the femur, which is more common in women. Abnormal control of lower limb movements and deficient neuromuscular parameters may lead to greater trochanteric pain syndrome; however, no studies have used neuromuscular training as a treatment strategy. Thus, this study aims to compare the effect of a protocol of general exercises versus a program of motor control training on pain at baseline and after treatment in women with greater trochanteric pain syndrome.

## Methods

The study was approved by the Research Ethics Committee (CAAE: 87372318.1.0000.5406) and has been prospectively registered on the Brazilian Registry of Clinical Trials (RBR-37gw2x). Sixty participants will be randomized to receive motor control exercises or general exercises. The application will be performed twice a week for 8 weeks. The participants will be evaluated before the treatment (T0), after 8 weeks of intervention (T8) and after 60 weeks of intervention (T60). The primary outcome measures will be the hip pain intensity, and secondary outcomes will be muscle strength, kinesiophobia, global perceived effect, pain catastrophization, central sensitization and quality of life.

**Data Availability Statement:** Research data will be made publicly available when the study is completed and published.

**Funding:** GTAN (Finance Code 001): Coordenação de Aperfeiçoamento de Pessoal de Nível Superior, Brasil (CAPES) - Doctorate's degree scholarship. https://www.gov.br/capes/pt-br The funders had and will not have a role in study design, data collection and analysis, decision to publish, or preparation of the manuscript.

**Competing interests:** The authors have declared that no competing interests exist.

## Conclusions

Studies have suggested that greater trochanteric pain syndrome may be related to poor hip and pelvic control, however, no study has investigated an exercise protocol focused on increasing the strength of the abductor and extensor muscles of the hip associated with pelvic control training, especially in positions of unilateral support, such as gait. This study will help determine whether greater trochanteric pain syndrome is related to abnormal control of lower limb movements.

## Background

Greater trochanteric pain syndrome (GTPS) is an overarching term used to define pain and tenderness in the greater trochanteric region of the femur [1, 2]. It is estimated that 10 to 25% of the population will develop some type of pain in the lateral hip region[1–3], predominantly in women over 40 years old [2–6]. GTPS directly impacts the worsening quality of life, which is similar to the findings in individuals with severe hip osteoarthritis [7]. Women present pathomechanisms that can influence the onset of the GTPS, such as, smaller gluteal tendinous insertions on the femur, shorter gluteal moment arm and weakness of the gluteus medius [6].

Changes in muscle function of the gluteus medius and minimus muscles may lead to poor hip and pelvic control, and leading to increased hip adduction in the frontal plane, especially in positions of unilateral support, such as single leg standing [6, 8]. Physical exercise are preferable to passive approaches in the treatment of GTPS as they result in better short-term [9, 10], medium [9] and long-term [9–11] outcomes in terms of pain and function when compared to rest, shockwave therapy and / or invasive treatments, e.g. corticosteroid injection (CSI) [5, 9, 11].

Individuals with GTPS have abnormal control of lower limb movements and deficient neuromuscular parameters [8, 12, 13] and alterations in trunk and pelvic kinematics during walking [14], however, no studies have used neuromuscular training as a treatment strategy and there is insufficient evidence about the influence of this intervention in terms of the clinical and biomechanical aspects. Thus, this study aims to compare the effect of a protocol of general exercises versus a program of motor control training on pain at baseline and after treatment, at 8 and 60 weeks in women with GTPS. We hypothesize that both the women undergoing the motor control protocol and the women undergoing the general exercise group (GEG) will show improvements in all evaluated outcomes, although the motor control group (MCG) will be superior due to the intervention being specific to the hip region.

## Material and methods

### Design

A two-arm, parallel randomized, double-blinded (outcome assessor and statistician), will be performed. All personal data will be confidential. The study follows the TIDieR (Template for Intervention Description and Replication) checklist [15] and the 2013 Standard Protocol Items: Recommendations for International Trials statement [16] (Fig 1).

### Ethical aspects

The study was approved by the Research Ethics Committee of the São Paulo State University of the Faculty of Philosophy and Sciences / Campus of Marília (CAAE: 87372318.1.0000.5406), first approval was in April, 20[th] 2018.

| | STUDY PERIOD | | | |
| --- | --- | --- | --- | --- |
| | Screening | Baseline | Post-allocation | |
| TIMEPOINT | $-t_1$ | 0 | Week 8 | Week 60 |
| **ENROLMENT:** | | | | |
| **Eligibility screen** | X | | | |
| **Informed consent** | X | | | |
| **Allocation** | | X | | |
| **INTERVENTIONS:** | | | | |
| *Motor control exercises* | | | ●━━━━━━━━━━● | |
| *General exercises* | | | ●━━━━━━━━━━● | |
| **ASSESSMENTS:** | | | | |
| *Pain* | | X | X | X |
| *Global perceived effect* | | | X | X |
| *Muscle strength* | | X | X | |
| *Pain catastrophization* | | X | X | X |
| *Kinesiophobia* | | X | X | X |
| *Central sensitization* | | X | X | X |
| *Quality of life* | | X | X | X |

**Fig 1. Schedule of the study protocol according to the standard protocol items: Recommendations for interventional trials checklist.**

The protocol of this study has been prospectively registered on the Brazilian Registry of Clinical Trials (RBR-37gw2x). Participants will be informed about all study procedures and asked to sign the Informed Consent Form prior to their enrollment in the study.

## Participants

**Recruitment.** Women with GTPS will be recruited in the community, universities and within the public health service, through digital (facebook and instagram) and written dissemination (flyers) placed in the public health service and at the universities. Interested participants will contact the main researcher via phone or social media.

## Eligibility criteria

**Inclusion and exclusion criteria.** Initial screening will be made by telephone. Potential participants will be eligible for the study if they are between 18 and 70 years old, do only recreational physical activity less than twice a week, and have had lateral hip pain for $\geq$ 3 months. They should have been feeling pain during one or more of these seven daily activities: walking, prolonged standing, standing up from the seated position, prolonged sitting, climbing stairs, climbing up hills or slopes, and side-lying. Potential participants will be ineligible for the study

if they have a body mass index (BMI) > 36kg / m², have received some type of invasive intervention for lateral hip pain (e.g. CSI) or physical therapy intervention for their hip pain in the last 12 weeks, have morning stiffness in the hip ≤ 60 minutes, any disease affecting the neuromuscular system or that may prevent data collection, have had spinal or hip surgery, any infectious condition, any neoplasm or cannot commit to participating for the duration of treatment [10, 11, 14, 17, 18].

## Procedures

Potential participants who fulfil the eligibility criteria by phone will then undergo two assessments. The first one will be a clinical screening and will be carried out to screen the inclusion and exclusion criteria that were not possible to evaluate by phone, as well as describing in more detail the nature of the study and commitment required, with the opportunity for the potential participant to ask questions. The clinical screening will be carried out at São Paulo State University and will take approximately 40 minutes to 1 hour. A registered physiotherapist with 8 years of experience, and who has a Master in musculoskeletal disorders will make the evaluation. Anthropometric data (age, weight, height, and BMI), personal information (name, address, telephone number), demographic partner (profession, race and education) and pathological and clinical history will be collected. The personal data of the participants will be numerically coded and stored in a database. It will be determined whether the participants have the ability to manipulate shoes and socks [19], have limitation in the range of motion of spine, hip or lower limb that affects gait or the data collection or visible lower limb discrepancy. It will be assessed whether the participants have palpation tenderness over the greater trochanteric region of the femur [8, 20, 21]. If both hips are symptomatic, the most painful side will be evaluated [10]. Trendelenburg sign, which is the pelvic drop during single leg stance, indicating inefficiency of hip abductor muscles, will be evaluated [5, 8, 20].

The tests that will be used for clinical evaluation are intended to define the exact location of the pain and to transmit compressive and/or tensile forces on the tendons of the abductor gluteal muscles over the greater trochanter. The presence of pain on palpation is an inclusion criteria, therefore, it is mandatory for participation in the study.

- Palpation of the greater trochanter: positive palpation tenderness, considered when there is pain over the greater trochanter of the femur, during the evaluation in lateral decubitus (LD) with the pain side up, flexion of the hip joint and knees together [5, 6, 8, 11, 18].

Because of the little benefit of the greater trochanteric palpation when the test is positive, due to the poor specificity, another test with high specificity will be used to rule in the greater trochanteric pain syndrome, to reduce false positive. Additionally to the positive greater trochanteric palpation test, the participants also must experience pain in one or more of the six provocative clinical pain tests [13, 17, 18]. They are:

- Hip FADER: Participant in supine position (SP), has to report pain over the greater trochanter when the affected lower limb is positioned in 90º flexion, adduction and external hip rotation [18].

- Hip FADER-R with static muscle test: Participant in SP, has to report pain when the affected lower limb is positioned in the same way but maintaining isometric resistance to internal rotation at the end of range of motion [5, 6, 8, 11, 18].

- Patrick-Faber test: Participant in SP, has to report pain when the affected lower limb is positioned in flexion, abduction and external hip rotation [6, 18, 19].

- ADD: Participant has to report pain when the affected lower limb is positioned in passive hip adduction in LD [18].

- ADD R: Participant has to report pain when the affected lower limb is positioned in passive hip adduction in LD but maintaining isometric resistance to abduction [18].

- Single leg stance (SLS): Participant has to report pain while standing on one leg for 30 seconds [6].

The second assessment will be carried out by two different blinded assessors, both registered physiotherapists with PhD in musculoskeletal disorders, to evaluate the participants outcomes. They are pain intensity, global perceived effect, muscle strength, pain catastrophization, kinesiophobia, central sensitization and quality of life. The blinded assessors will evaluate the participants at baseline (before the treatment), after 8 weeks of intervention and after 60 weeks of intervention.

## Research team

The trial will involve five researchers; two researcher responsible for evaluations; one researcher responsible for interventions; one researcher responsible for randomising participants and one researcher who will perform the statistical analysis.

## Sample size and power analysis

The sample size calculation was performed through the G $^*$ Power software and was based on the ability to detect a difference of 2 points (pilot standard deviation of ±0,8 points) of pain intensity score, because it is the primary outcome of the research. A power of 0.80, probability of error α 0.05, effect size of 0,48 and a 15% dropout rate were used, thus we will recruit 60 participants (30 in each group) [9].

## Evaluations

The participants will be evaluated before the treatment (T0), after 8 weeks of intervention (T8) and after 60 weeks of intervention (T60).

## Randomization, allocation and blinding

If all eligibility criteria are fulfilled, and if the presence of GTPS is determined, the participants will be randomly assigned to receive either 1) motor control exercises or 2) general exercises (Fig 2). At this point, before the blind evaluation, participants will give their free and informed consent.

Participants will be randomly allocated to two therapeutic arms using permuted, block-randomization to balance the number of participants allocated to each group. The permuted block (with six participants per block) randomization sequence will be generated by the website www.sealedenvelope.com. The participants will be informed of their random allocation by one of the researchers not involved with the assessment process. After the randomization, the participants will be invited to return for physiotherapeutic evaluation (T0), with blinded assessors both registered physiotherapists with PhD in musculoskeletal disorders, to evaluate the participants outcomes. These assessors will not participate in the screening or interventions. All participants will be advised not to disclose to the assessor any details about the intervention program that they have received (to ensure that allocation is concealed from the assessor).

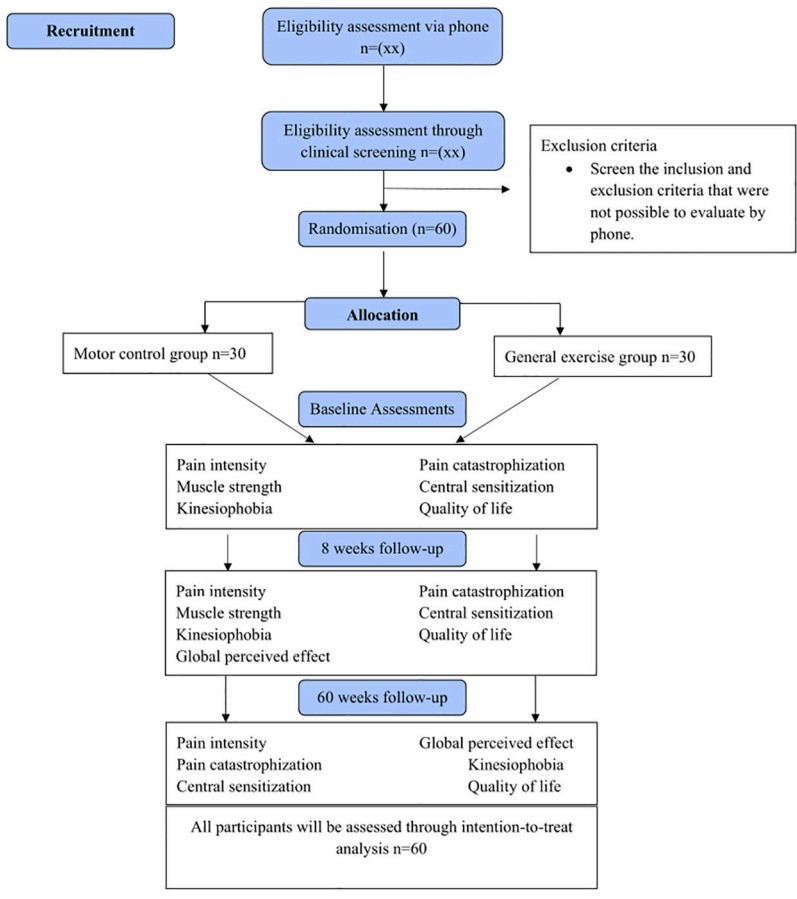

**Fig 2. Study fluxogram.**

## Outcome measures

**Primary outcome measure.** There is one primary outcomes measure: 1) Average pain over the previous week.

1) Pain intensity will be assessed at baseline (T0) and after treatment at 8 weeks (T8). It will also be assessed at 60 weeks (T60), which is a secondary time point for the primary outcome. Pain intensity will be assessed by the visual analog scale (VAS) [5]. Formed by a horizontal line of 100 mm, anchored by the words "no pain / discomfort" and "worst pain / discomfort imaginable" [5]. To measure the pain, the participant will be asked to indicate the level pain they are experiencing at the present moment and the average pain they experienced in the last week by marking the scale with a line. A ruler will be used to measure the value obtained [5].

**Secondary outcomes measures.** There are five secondary outcomes measures: 1) Global perceived effect, 2) Muscle strength, 3) Pain catastrophization, 4) Kinesiophobia, 5) Central sensitization and 6) Quality of life.

1. The scale of global perceived effect (GPE) will be used after treatment, at 8 (T8) and 60 weeks (T60) during post-intervention reassessment [22]. This is a 7-point scale (1 = completely recovered, 7 = worse than ever) to assess recovery. GPE evaluates the participant's perception regarding the modification of their clinical condition after the intervention, and is assessed by a simple and easily understood question with alternative answer options which will be dichotomized into "improved" ("completely recovered" and "much

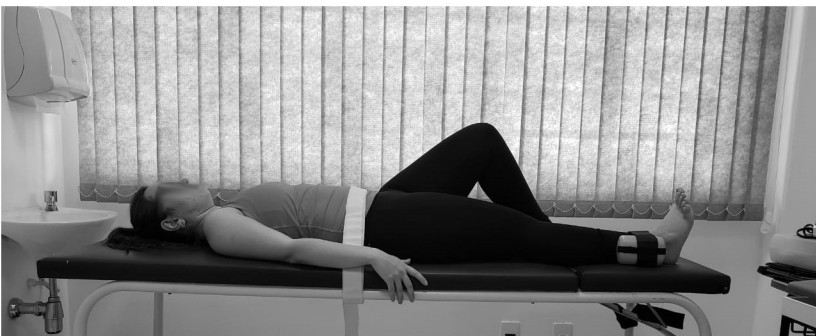

**Fig 3. Hip abductor muscles evaluation.**

improved") versus "not improved" ("slightly improved," "not changed," "slightly worsened," "much worsened," "worse than ever" [22].

2. The isometric strength of the hip abductor and extensor muscles will be assessed at baseline (T0) and after treatment, at 8 weeks (T8) during post-intervention reassessment. A Lafayette Manual Muscle Tester (Lafayette Instruments) will be used, which has been shown to be a valid method of assessing isolated muscle contraction strength [23]. To evaluate the hip abductor muscles, the participants will be positioned supine on a stretcher, stabilized by velcro bands around the pelvis and above the lateral ankle malleolus to avoid compensatory movements and the influence of the examiner's resistance [23]. The hand dynamometer will be positioned above the lateral ankle malleolus and fixed to the stretcher by an inelastic band, it will be positioned without rotation, with 10° of hip abduction to minimize the compression potential of the tendons against the greater trochanter [13]. The untested lower limb will be positioned with 45° of hip and knee flexion, with the foot positioned on the stretcher and the upper limbs resting at the side of the body (Fig 3) [13].

To evaluate the hip extensors, the participants will be positioned in the prone position (PP) with knee flexion in the lower limb to be evaluated. The hand dynamometer will be positioned on the posterior thigh, above the popliteal fossa and will be stabilized by an inelastic velcro band [24]. Another Velcro band will be positioned around the pelvis (Fig 4) [24]. Measurement of limb lengths will be taken with a standard cloth tape measure in order to calculate torque, from the greater trochanter to the center of the dynamometer, and will be recorded in meters [24, 25]. The tests of maximal isometric strength will be performed 3

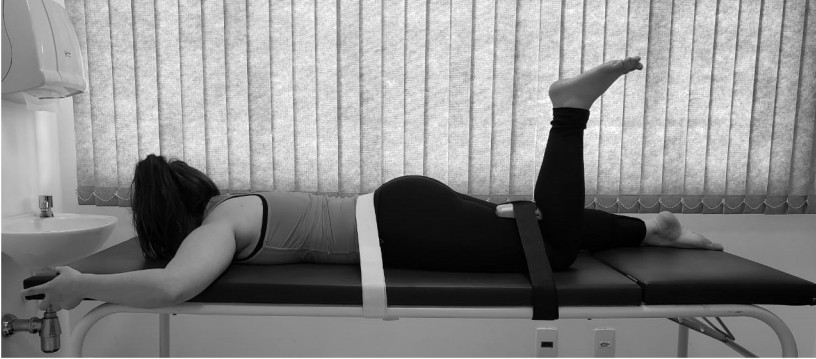

**Fig 4. Hip extensor muscles evaluation.**

times with 5 seconds duration for each contraction, with 30 seconds of rest between each attempt [24]. Participants will be verbally encouraged to perform as much force as possible during the test. Strength values will be normalized by the weight of each participant [24].

3. Pain catastrophization will be measured with the Pain Catastrophizing Scale (PCS) at baseline (T0) and after treatment, at 8 (T8) and 60 weeks (T60) during post-intervention reassessment. PSC is a validated [26], self-administered, 13-item questionnaire that assesses catastrophic thoughts, feelings, and behavior when in pain [26]. This questionnaire assesses three major domains: helplessness, magnification and rumination in relation to pain. Results are calculated by summing all survey items and total scores range from 0 to 52, with higher scores indicating higher levels of pain catastrophization [26].

4. Kinesiophobia will be assessed with the Tampa Scale for Kinesiophobia (TSK), which is a validated, 17 question, self-administered questionnaire which has been translated into Portuguese. Kinesiophobia will be assessed at baseline (T0) and after treatment, at 8 (T8) and 60 weeks (T60) during post-intervention reassessment. The final score can be at least 17 and at most 68 points, with higher scores indicating higher degrees of kinesiophobia [27].

5. Central sensitization will be measured with the central sensitization inventory, which is a self-reported health symptoms questionnaire designed as an easy-to-administer tracker for patients who are at high risk for central sensitization, or for assessing symptoms related to central sensitization [28]. Central sensitization will be measured at baseline (T0) and after treatment, at 8 (T8) and 60 weeks (T60) during post-intervention reassessment. It has also been recommended as a component of an algorithm to help classify chronic pain patients with central sensitization and to help differentiate them from patients with primary neuropathic and nociceptive pain [28].

6. The quality of life will be assessed using the International Hip Outcome Tool (iHOT) which is a 12-question self-administered questionnaire. The questions are evaluated using a visual analogue scale, so each question has a 10 cm line and the participants must add a vertical line crossing the horizontal line, and the farther to the left, the worse the symptoms. The result of each question can vary between 0 and 100 and to determine the result, all the questions must be added and divided by the number of answered questions [29]. The quality of life will be measured at baseline (T0) and after treatment, at 8 (T8) and 60 weeks (T60) during post-intervention reassessment.

## Interventions

**Motor control group.** The motor control program will be performed over 8 weeks, with two weekly appointments, face-to-face and individualized. There will be a total of 16 appointments, each lasting 50–60 minutes. The protocol will consist of isotonic and isometric strengthening exercises, focused on abductor and extensor muscles of the hip with coordination through verbal commands to improve the dynamic motor control of the lower limbs. Exercise progression will occur through elastic bands (Domyos), from the easiest to the most difficult, and the addition of more difficult exercises [24]. In the first week there will be no use of load, however from the second week the MCG will be tested with three different colors of elastic resistance bands (green = easy, blue = medium and orange = hard), from the easiest to the hardest and will be instructed to perform 3–5 repetitions with each band [24]. They will decide which elastic band they feel they would be able to perform 3 sets with 8–12 repetitions, maintaining the quality of movement [24]. Participants will choose the color of elastic band

for each exercise and the elastic band will be positioned above the knee joint. The load progression test will be performed weekly and the load progression will be increased according to the modified Borg scale (0–10), when 3 (moderate) or lower scores (easy) are reached, the load progression will be made to the subsequent elastic band [24]. They will also use an aerobic step in the last two weeks. The rest between exercises will be one minute [30]. As for the evolution of the exercises, if the participants are unable to progress due to pain enough to give up the exercise, or difficulty in performing at least 8 repetitions with quality, they will remain with the parameters they were able to perform (Table 1).

**Table 1. Exercise dosage and progressions of motor control protocol.**

| Weeks | Exercise | Sets | Reps | Position | Material |
|---|---|---|---|---|---|
| 1st and 2nd | Bilateral bridge | 3 | Hold 15s / 8–12 | Supine | Elastic band |
| | Bridge with feet together | 3 | Hold 15s / 8–12 | Supine | Elastic band |
| | Clamshell | 3 | Hold 15s / 8–12 | Lateral decubitus | Elastic band |
| | Hip abduction with lower limbs flexed | 3 | Hold 15s / 8–12 | Lateral decubitus | Elastic band |
| | Double leg squat | 3 | Hold 15s / 8–12 | Standing | Elastic band |
| 3rd and 4th | Single leg bridge | 3 | Hold 15s / 8–12 | Supine | Elastic band |
| | Hip abduction with lower limbs extended | 3 | Hold 15s / 8–12 | Lateral decubitus | Elastic band |
| | Hip and knee extension | 3 | Hold 15s / 8–12 | Four-point | Elastic band |
| | Hip abduction | 3 | Hold 15s / 8–12 | Standing | Elastic band |
| | Double leg squat | 3 | Hold 15s / 8–12 | Standing | Elastic band |
| 5th and 6th | Single leg bridge with hip abduction | 3 | 8–12 | Supine | Elastic band |
| | Hip abduction, flexion and extension | 3 | 8–12 | Lateral decubitus | Elastic band |
| | Hip abduction | 3 | Hold 15s | Four-point | Elastic band |
| | Hip abduction and extension | 3 | 8–12 | Four-point | Elastic band |
| | Side walking | 3 | 8–12 | Standing | Elastic band |
| | Reverse lunge | 3 | 8–12 | Standing | Elastic band |
| | Single leg stance | 3 | Hold 15s | Standing | Elastic band |
| | Single leg stance and squat | 3 | 8–12 | Standing | Elastic band |
| 7th and 8th | Side walking | 3 | 8–12 | Standing | Elastic band |
| | Side walking with squat | 3 | 8–12 | Standing | Elastic band |
| | Single leg stance with trunk rotation | 3 | 8–12 | Standing | Elastic band |
| | Single leg stance with flexion and extension of the contralateral hip | 3 | 8–12 | Standing | Elastic band |
| | Single leg stance with flexion, abduction and extension of the hip | 3 | 8–12 | Standing | Elastic band |
| | Hip Hike | 3 | Hold 15s / 8–12 | Standing | Elastic band Step |
| | Walk | 3 | 8–12 | Standing | Elastic band |

Reps: Repetitions; s: Seconds

**General exercise group.** The general exercise program will be performed over 8 weeks, with two weekly appointments, face-to-face and individualized. There will be a total of 16 appointments, each lasting 50–60 minutes. The protocol will consist of 5 minutes walking warm-up, stretching and strengthening of the muscle groups of the lower trunk, hip and lower limbs. Exercise progression will occur through elastic bands (Domyos), from the easiest to the most difficult (green = easy, blue = medium and orange = hard), with the use of an exercise ball and the addition of more difficult exercises [24]. In the first week there will be no use of load, however from the second week the GEG will be tested with three different colors of elastic resistance band, from the easiest to the hardest and instructed to perform 3–5 repetitions with each band [24]. They will decide which elastic band they feel they would be able to perform 3 sets with 8–12 repetitions, whilst maintaining the quality of movement [24]. Participants will choose the color of elastic band for each exercise, and the elastic band will be positioned above the knee joint. The load progression test will be performed weekly and the load progression will be increased according to the modified Borg scale (0–10), when 3(moderate) or lower scores (easy) are reached, the load progression will be made to the subsequent elastic band [24]. The rest between exercises will be one minute [30]. As for the evolution of the exercises, if the participants are unable to progress due to pain enough to give up the exercise, or difficulty in performing at least 8 repetitions with quality, they will remain with the parameters they were able to perform (Table 2).

## Statistical analysis

The study will be run as a superiority trial. The statistical analysis will follow the intention-to-treat concept. Statistical analysis will be performed using the software IBM SPSS Statistics for Windows, version 20.0 (IBM Corp., Armonk, N.Y., USA) and statistician will be blind. Data will be evaluated using exploratory statistical techniques. Firstly, the normality and homogeneity of the data will be verified and then the appropriate statistical analyses will be adopted for the variables. The between-group differences for the primary and secondary outcomes and their respective confidence intervals at 95% will be calculated by constructing mixed linear models. "Time" and "group" will be considered fixed effects, whereas the participants will be considered the random effect. The time by group interaction will be included in the analysis to assess the difference effect between the groups at each follow-up, and the dependent variable baseline value will be included as a covariate for the correction of possible differences [31]. The significance level will be 0.05 for all statistical analyses.

A causal mediation analysis will be conducted using the "mediate" package in R (The R Foundation for Statistical Computing). A model-based inference approach will be used to estimate the average causal mediation effect (ACME), average direct effect (ADE) and the average total effect for pain [32]. Two regression models will be created to each outcome: the mediator model and the outcome model. If there is no total effect, we will conduct several univariate mediation models to verify where the causal pathway braked down. By the other hand, if there is a between-group difference, we will construct a multivariate mediation model to verify how the intervention works through the putative mediators.

The mediator model will be constructed with treatment allocation as the independent variable and the putative mediator as the dependent variable. The outcome model will be constructed with the treatment allocation and the putative mediator as independent variables and the outcome as independent variable. The outcome models were adjusted for potential confounders (i.e. age, sex). Continuous mediators and outcomes normally distributed will be modelled using linear models (*lm*).

**Table 2. Exercise dosage and progression of general exercises.**

| Weeks | Exercises | Sets | Reps | Position | Material |
|---|---|---|---|---|---|
| 1st and 2nd | Warm up walk | 1 | 5 minutes | | |
| | Unilateral hamstring stretch | 1 | Hold 60s | Supine | |
| | Bilateral hamstring stretch | 1 | Hold 60s | Supine | |
| | Hip external rotation mobility | 3 | 8–12 | Supine | |
| | Hip internal rotation mobility | 3 | 8–12 | Supine | |
| | Hip flexors strengthening with flexed lower limbs | 3 | 8–12 | Supine | Elastic band |
| | Hip adductors strengthening with knees flexed | 3 | 8–12 | Supine | Ball |
| | Hip extensors strengthening with knees flexed | 3 | 8–12 | Prone | Elastic band |
| | Knee extensors stretching | 1 | Hold 60s | Prone | |
| | Hip adductors stretching | 1 | Hold 60s | Sitting down | |
| 3rd and 4th | Warm up walk | 1 | 5 minutes | | |
| | Piriformis stretching | 1 | Hold 60s | Supine | |
| | Abductor stretching | 1 | Hold 60s | Supine | |
| | Hip flexors strengthening with one leg straight moving up and down | 3 | 8–12 | Supine | Elastic band |
| | Hip extensors strengthening with the lower limbs extended | 3 | 8–12 | Prone | Elastic band |
| | Hip adductors strengthening with knees flexed | 3 | 8–12 | Supine | Ball |
| | Hip abductors strengthening with knees flexed | 3 | 8–12 | Supine | Elastic band |
| | Hip adductors stretching | 1 | Hold 60s | Sitting | |
| | Hip flexors stretching | 1 | Hold 60s | Semi-sitting | |
| | Knee extensors stretching | 1 | Hold 60s | Standing | |
| 5th and 6th | Warm up walk | 1 | 5 minutes | | |
| | Trunk flexor stretching with elbows supported | 1 | Hold 60s | Prone | |
| | Hip flexors strengthening with trunk elevated | 3 | 8–12 | Supine | Elastic band |
| | Adductor muscles stretching | 1 | Hold 60s | Sitting down | |
| | Trunk extensor stretching | 1 | Hold 60s | Sitting down | |
| | Sitting on the feet, stretching the body forward | 1 | Hold 60s | | |
| | Trunk lateral stretching | 1 | Hold 60s | Sitting down | |
| | Hip adductors strengthening | 3 | 8–12 | Standing | |
| | Squats against wall | 3 | 8–12 | Standing | Ball |
| | Hip abductor strengthening | 3 | 8–12 | Standing | Elastic band |
| | Knee extensors stretching | 1 | Hold 60s | Standing | |
| 7th and 8th | Warm up walk | 1 | 5 minutes | | |
| | Trunk flexor stretching with upper limbs straight | 1 | Hold 60s | Prone | |
| | Trunk and hip extensors strengthening elevating upper and lower limbs | 3 | 8–12 | Prone | |
| | Trunk rotation stretching | 1 | Hold 60s | Sitting down | |
| | Trunk strengthening by flexing arm and contralateral leg. | 3 | 8–12 | four-point stance | Elastic band |
| | Sitting on the feet, stretching the body forward | 1 | Hold 60s | | |
| | Hamstring strengthening | 3 | 8–12 | Standing | Elastic band |
| | Hip flexors strengthening | 3 | 8–12 | Standing | Elastic band |
| | Hip extensors strengthening | 3 | 8–12 | Standing | Elastic band |
| | Iliotibial band stretching | 1 | Hold 60s | Standing | |

Reps: Repetitions; s: Seconds

The *mediate* function will be used to estimate the value of the mediator and outcome. The simulated potential values of the mediator and outcome will be used to compute the ACME, ADE and average total effect. We will use 1000 bootstrap simulations to generate 95% confidence intervals (95% CI) if linear assumptions of mediator and/or outcome models are not violated. Non-parametric bootstrap simulations will be used if the linear assumptions of the mediator and/or outcome models are violated. The putative mediators that will be investigated in both models will be hip extensors and abductors muscle strength.

## Discussion

This study aims to evaluate whether there will be a difference in pain intensity between individuals with GTPS who have undergone an exercise protocol with an emphasis on motor control training and those who have undergone a program of general, nonspecific exercises. Taking into consideration that no studies have used neuromuscular training as a treatment strategy for GTPS, and there is insufficient evidence about the influence of this intervention in terms of the clinical and biomechanical aspects, this study is necessary.

Exercises are considered to be the cornerstone of non-surgical treatment for chronic musculoskeletal pain [18]. We chose two types of protocols: the MCG was developed to be specific to the targeted muscle group, focusing not only on strengthening, but also on improving gait patterns, maintaining correct, efficient movement patterns, and providing guidance on how to avoid aggravating positions such as excessive adduction of the femur during functional activities. For GEG, a more general, non-specific exercise protocol will be applied, without advice on positioning or strategies to avoid aggravation of pain–only warm-up, stretching and non-specific strengthening exercises will be performed.

The diagnosis of GTPS is clinical and two studies by Grimaldi et al. 2017 and Ganderton et al. 2017 demonstrated that palpation over the greater trochanter has a sensitivity of 80% and 85.7% respectively, and a specificity of 47% and 61.1% respectively. They also reported likelihood ratios of 0.43 and 2.2, respectively, demonstrating that this test alone, when negative, is able to rule out the presence of GTPS [6, 20]. However, the combination of palpation and one other of the FABER, SLS, FADER, FADER-R, ADD, ADD-R tests increases the chances of detecting GTPS due to the specificity of these tests, which can reach 100%, for this reason, we will apply a combination of multiple clinical tests to avoid the inclusion of participants who do not have GTPS [5, 6, 18, 20].

The clinical trial will conform to the standards of the consort group [33]. Study participants will be randomized to participate in one of the groups through concealed allocation [18, 33]. It is not possible to blind the clinician or the participant to group allocation. Only the assessor of the outcome measures and the statistician are able to be blinded to treatment allocation. The importance of not disclosing anything to this assessor about the nature of their treatment will be strongly emphasized to participants. The statistical analysis will be conducted blind to treatment group allocation–the actual groups will only be revealed post analysis. In addition, intention-to-treat analysis will be used [34]. As a limitation of the study, we highlight the participation of women only, so care must be taken when extrapolating data to men with the same conditions.

The idea of the research arose due to studies that pointed out physical exercises as the first line of treatment for the management of tendinopathies [5, 17, 18]. General exercise has the ability to decrease nociceptive afferent input to the central nervous system (CNS) and thus decrease pain [10], however our motor control protocol consists, in addition to isotonic exercises, of isometric exercises that have been reported to target both peripheral and central pain drivers by releasing cortical inhibition and reducing tendon pain [35]. In addition, the MCG

consists of neuromuscular training, which may be the reason for this protocol to be superior to the GEG. The findings of this study will contribute to determining the effect on pain of both MCG and GEG in the management of individuals with GTPS. This information may be used by health professionals to assist them in clinical decision making and selecting the most appropriate training program for management of GTPS.

## Trial status

Participants will be recruited and will receive treatment until 2022.

## Supporting information

**S1 Checklist. SPIRIT 2013 checklist: Recommended items to address in a clinical trial protocol and related documents**∗.
(PDF)

**S1 Protocol.**
(DOCX)

**S2 Protocol.**
(DOCX)

**S3 Protocol.**
(PDF)

## Author Contributions

**Conceptualization:** Guilherme Thomaz de Aquino Nava, Caroline Baldini Prudencio, Rafael Krasic Alaiti, Beatriz Mendes Tozim, Cristiane Rodrigues Pedroni, Angélica Mércia Pascon Barbosa, Marcelo Tavella Navega.

**Data curation:** Guilherme Thomaz de Aquino Nava, Rafael Krasic Alaiti.

**Formal analysis:** Rafael Krasic Alaiti.

**Funding acquisition:** Guilherme Thomaz de Aquino Nava.

**Investigation:** Guilherme Thomaz de Aquino Nava, Caroline Baldini Prudencio, Beatriz Mendes Tozim, Rebecca Mellor, Cristiane Rodrigues Pedroni, Angélica Mércia Pascon Barbosa, Marcelo Tavella Navega.

**Methodology:** Guilherme Thomaz de Aquino Nava, Caroline Baldini Prudencio, Rafael Krasic Alaiti, Beatriz Mendes Tozim, Rebecca Mellor, Cristiane Rodrigues Pedroni, Angélica Mércia Pascon Barbosa, Marcelo Tavella Navega.

**Project administration:** Guilherme Thomaz de Aquino Nava, Marcelo Tavella Navega.

**Software:** Guilherme Thomaz de Aquino Nava, Caroline Baldini Prudencio, Beatriz Mendes Tozim, Cristiane Rodrigues Pedroni, Angélica Mércia Pascon Barbosa, Marcelo Tavella Navega.

**Supervision:** Guilherme Thomaz de Aquino Nava, Marcelo Tavella Navega.

**Validation:** Guilherme Thomaz de Aquino Nava, Caroline Baldini Prudencio, Rebecca Mellor, Cristiane Rodrigues Pedroni, Angélica Mércia Pascon Barbosa, Marcelo Tavella Navega.

**Visualization:** Guilherme Thomaz de Aquino Nava, Marcelo Tavella Navega.

**Writing – original draft:** Guilherme Thomaz de Aquino Nava, Caroline Baldini Prudencio, Rafael Krasic Alaiti, Beatriz Mendes Tozim, Rebecca Mellor, Cristiane Rodrigues Pedroni, Angélica Mércia Pascon Barbosa, Marcelo Tavella Navega.

**Writing – review & editing:** Guilherme Thomaz de Aquino Nava, Caroline Baldini Prudencio, Rafael Krasic Alaiti, Beatriz Mendes Tozim, Rebecca Mellor, Cristiane Rodrigues Pedroni, Angélica Mércia Pascon Barbosa, Marcelo Tavella Navega.

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
