## [Decision Letter · Decision Letter 0]

1 Apr 2022

PONE-D-21-22065

Motor control exercises versus general exercises for greater trochanteric pain syndrome: A protocol of a randomized controlled trial

PLOS ONE

Dear Dr. Thomaz de Aquino Nava,

Thank you for submitting your manuscript to PLOS ONE. After careful consideration, we feel that it has merit but does not fully meet PLOS ONE’s publication criteria as it currently stands. Therefore, we invite you to submit a revised version of the manuscript that addresses the points raised during the review process.

The manuscript has been evaluated by three reviewers, and their comments are available below.

The reviewers have raised a number of concerns that need attention. They request additional information regarding the planned consent method and clarification on the primary and/or secondary aspects for this study. The reviewers also request more information regarding the methodological aspects of the study (such as the planned statistical analysis)

Could you please revise the manuscript to carefully address the concerns raised?

We look forward to receiving your revised manuscript.

Kind regards,

Jamie Royle, PhD

Associate Editor

PLOS ONE

“This study was financed by the Coordenação de Aperfeiçoamento de Pessoal de Nível Superior, Brasil (CAPES)—Doctorate’s degree scholarship, Finance Code 001.”

“GTAN (Finance Code 001): Coordenação de Aperfeiçoamento de Pessoal de Nível Superior, Brasil (CAPES) - Doctorate’s degree scholarship.

https://www.gov.br/capes/pt-br

The funders had and will not have a role in study design, data collection and analysis, decision to publish, or preparation of the manuscript.”

Reviewers' comments:

Reviewer's Responses to Questions

**Comments to the Author**

1. Does the manuscript provide a valid rationale for the proposed study, with clearly identified and justified research questions?

Reviewer #1: Yes

Reviewer #2: Yes

Reviewer #3: Yes

2. Is the protocol technically sound and planned in a manner that will lead to a meaningful outcome and allow testing the stated hypotheses?

Reviewer #1: Yes

Reviewer #2: Yes

Reviewer #3: Yes

3. Is the methodology feasible and described in sufficient detail to allow the work to be replicable?

Reviewer #1: Yes

Reviewer #2: Yes

Reviewer #3: Yes

4. Have the authors described where all data underlying the findings will be made available when the study is complete?

Reviewer #1: No

Reviewer #2: Yes

Reviewer #3: Yes

5. Is the manuscript presented in an intelligible fashion and written in standard English?

Reviewer #1: Yes

Reviewer #2: Yes

Reviewer #3: Yes

6. Review Comments to the Author

You may also provide optional suggestions and comments to authors that they might find helpful in planning their study.

Reviewer #1: Motor control exercises versus general exercises for greater trochanteric pain syndrome: A protocol of a randomized controlled trial

-I find this a very interesting topic, thank you for addressing the pain of the greater trochanteric region in this study.

-I am going to make a series of recommendations to make them clearer for the reader.

-It is advisable not to use abbreviations in the abstract.

-Line 146- when they talk "the second one will be carried out by a blinded-assessor", it is not clear what this assessor is doing in this sentence. In fact it is not clear from the manuscript what this evaluator does. Please clarify what tasks this evaluator does in the manuscript, as it is not clear, nor how many times he/she evaluates the subject (line 211).

-At what point do the participants sign the informed consent, according to the Randomization, allocation and blinding section, is when they are going to receive the intervention program. The informed consent should be signed at the moment that the subjects want to participate once they have been thoroughly explained what the study consists of.

-Review bibliography number 28, it does not follow the format.

I hope that with these clarifications you can publish your protocol.

Reviewer #2: The authors plan to enroll 60 participant to conduct a two-arm, parallel randomized, double-blinded trial to compare the effect of general exercises v.s. a motor control training on pain at baseline and after treatment in women with greater trochanteric pain syndrome (GTPS).

1. Line 197. “effect size of 0,5” Please clarify whether this is in standard deviation unit? If not, what standard deviation was used in this power analysis.

2. Line 370. “appropriate statistical analyses will be adopted for the variables”. Please be specific for “appropriate statistical analyses”.

3. Line 377. “A causal mediation analysis will be conducted..” please be specific what mediation analysis will be conducted.

Reviewer #3: 1.) I am pleased to see the VISA-G outcome tool is mentioned within this protocol to improve homogenous outcome measures within GTPS research. However, it is unclear why this is not included as an outcome within the protocol submitted after ethical approval of the original document. It is unclear if the VISA-G is to be used as an outcome measure or not. If it is, please add this into either the primary or secondary outcomes. If it is not to be used as an outcome measure, why not? This has been previously recommended to be used in GTPS research to improve outcome homogeneity.

2.) You have stated that greater tochanteric palpation in conjunction with one other test will be used. It would be informative to state that this is because GT palp is of little benefit when the test is positive due to the poor Sp. Therefore, in the absence of a negative GT palp test, another test with high Sp will be used to rule in the pathology of interest (GTPS) to reduce your false positive inclusion rate.

7. PLOS authors have the option to publish the peer review history of their article (what does this mean?). If published, this will include your full peer review and any attached files.

Reviewer #1: No

Reviewer #2: No

Reviewer #3: No

---

## [Author Response · Author response to Decision Letter 0]

13 Apr 2022

Reviewer #1

It is advisable not to use abbreviations in the abstract.

Thank you for the guidance, the change has been made.

Line 146- when they talk "the second one will be carried out by a blinded-assessor", it is not clear what this assessor is doing in this sentence. In fact it is not clear from the manuscript what this evaluator does. Please clarify what tasks this evaluator does in the manuscript, as it is not clear, nor how many times he/she evaluates the subject (line 211).

Thank you for the guidance, the change has been made in the manuscript. “The second assessment will be carried out by two different blinded assessors, both registered physiotherapists with PhD in musculoskeletal disorders, to evaluate the participants outcomes. They are pain intensity, global perceived effect, muscle strength, pain catastrophization, kinesiophobia, central sensitization and quality of life. The blinded assessors will evaluate the participants at baseline (before the treatment), after 8 weeks of intervention and after 60 weeks of intervention.”

At what point do the participants sign the informed consent, according to the Randomization, allocation and blinding section, is when they are going to receive the intervention program. The informed consent should be signed at the moment that the subjects want to participate once they have been thoroughly explained what the study consists of.

Thank you for the guidance, the change has been made. 

“Randomization, allocation and blinding: If all eligibility criteria are fulfilled, and if the presence of GTPS is determined, the participants will be randomly assigned to receive either 1) motor control exercises or 2) general exercises (fig 2). At this point, before the blind evaluation, participants will give their free and informed consent.”

Review bibliography number 28, it does not follow the format.

We revised the bibliography using mendeley software. We believe that the error in the bibliographic references has been corrected.

Reviewer #2

Line 197. “effect size of 0,5” Please clarify whether this is in standard deviation unit? If not, what standard deviation was used in this power analysis.

Thank you for the guidance, the change has been made. 

“The sample size calculation was performed through the G * Power software, and was based on the ability to detect a difference of 2 points (pilot standard deviation of ±0,8 points) of pain intensity score, because it is the primary outcome of the research. A power of 0.80, probability of error α 0.05, effect size of 0,48 and a 15% dropout rate were used, thus we will recruit 60 participants (30 in each group)”

Line 370. “appropriate statistical analyses will be adopted for the variables”. Please be specific for “appropriate statistical analyses”.

Thank you for the guidance, the change has been made.

Line 377. “A causal mediation analysis will be conducted..” please be specific what mediation analysis will be conducted.

Thank you for the guidance, the change has been made.

Reviewer #3

I am pleased to see the VISA-G outcome tool is mentioned within this protocol to improve homogenous outcome measures within GTPS research. However, it is unclear why this is not included as an outcome within the protocol submitted after ethical approval of the original document. It is unclear if the VISA-G is to be used as an outcome measure or not. If it is, please add this into either the primary or secondary outcomes. If it is not to be used as an outcome measure, why not? This has been previously recommended to be used in GTPS research to improve outcome homogeneity.

We appreciate the question and the opportunity to answer it. We understand the importance of using this questionnaire to assess the disability outcome. When we started the study, we discovered that the VISA-G questionnaire would be translated, culturally adapted, and validated for the population with greater trochanter pain syndrome, however, we had to start data collection before the publication of the article validating the VISA-G for the Brazilian Portuguese language, which only took place in early 2021.

Therefore, of the aforementioned questionnaires, we only used the International hip outcome tool. Unfortunately, we realized the error only after submission, we forgot to remove the information from the "ethical aspects", as well as, to add the assessment of the quality of life in the secondary outcomes.

You have stated that greater tochanteric palpation in conjunction with one other test will be used. It would be informative to state that this is because GT palp is of little benefit when the test is positive due to the poor Sp. Therefore, in the absence of a negative GT palp test, another test with high Sp will be used to rule in the pathology of interest (GTPS) to reduce your false positive inclusion rate.

Thank you for the guidance, the change has been made.

“The presence of pain on palpation is an inclusion criteria, therefore, it is mandatory for participation in the study.

• Palpation of the greater trochanter: positive palpation tenderness, considered when there is pain over the greater trochanter of the femur, during the evaluation in lateral decubitus (LD) with the pain side up, flexion of the hip joint and knees together[5,6,8,11,18].

Because of the little benefit of the greater trochanteric palpation when the test is positive, due to the poor specificity, another test with high specificity will be used to rule in the greater trochanteric pain syndrome, to reduce false positive. Additionally to the positive greater trochanteric palpation test, the participants also must experience pain in one or more of the six provocative clinical pain tests[13,17,18]. They are:”

Changes to the reference list

Due to the changes suggested by the reviewers, we have added 2 new references:

Reference 29, due to the addition of a study outcome.

Reference 31, due to changes made in the statistical analysis.

---

## [Decision Letter · Decision Letter 1]

18 May 2022

Motor control exercises versus general exercises for greater trochanteric pain syndrome: A protocol of a randomized controlled trial

PONE-D-21-22065R1

Dear Dr. Thomaz de Aquino Nava,

We’re pleased to inform you that your manuscript has been judged scientifically suitable for publication and will be formally accepted for publication once it meets all outstanding technical requirements.

Kind regards,

George Vousden

Staff Editor

PLOS ONE

Additional Editor Comments (optional):

Reviewers' comments:

Reviewer's Responses to Questions

**Comments to the Author**

1. Does the manuscript provide a valid rationale for the proposed study, with clearly identified and justified research questions?

Reviewer #1: Yes

Reviewer #2: Yes

Reviewer #3: Yes

2. Is the protocol technically sound and planned in a manner that will lead to a meaningful outcome and allow testing the stated hypotheses?

Reviewer #1: Yes

Reviewer #2: Yes

Reviewer #3: Yes

3. Is the methodology feasible and described in sufficient detail to allow the work to be replicable?

Reviewer #1: Yes

Reviewer #2: Yes

Reviewer #3: Yes

4. Have the authors described where all data underlying the findings will be made available when the study is complete?

Reviewer #1: Yes

Reviewer #2: Yes

Reviewer #3: Yes

5. Is the manuscript presented in an intelligible fashion and written in standard English?

Reviewer #1: Yes

Reviewer #2: Yes

Reviewer #3: Yes

6. Review Comments to the Author

You may also provide optional suggestions and comments to authors that they might find helpful in planning their study.

Reviewer #1: Thank you for correcting and adapting all the corrections in the manuscript. For my part I have no more appreciations to include in your manuscript to be published.

Best regards

Reviewer #2: Thank you. All previously raised comments have been successfully addressed. I have no further comments

Reviewer #3: Thank you for making the suggested changes to the manuscript. I have no further comments to make regarding the manuscript.

7. PLOS authors have the option to publish the peer review history of their article (what does this mean?). If published, this will include your full peer review and any attached files.

Reviewer #1: No

Reviewer #2: No

Reviewer #3: No

---

## [Editor Report · Acceptance letter]

14 Jun 2022

PONE-D-21-22065R1 

Motor control exercises versus general exercises for greater trochanteric pain syndrome: A protocol of a randomized controlled trial 

Dear Dr. Thomaz de Aquino Nava:

I'm pleased to inform you that your manuscript has been deemed suitable for publication in PLOS ONE. Congratulations! Your manuscript is now with our production department. 

Kind regards, 

on behalf of

Dr. George Vousden 

Staff Editor

PLOS ONE